# Mechanisms of redundancy and specificity of the *Aspergillus fumigatus* Crh transglycosylases

Wenxia Fang[1,2], Ana Belén Sanz[3], Sergio Galan Bartual[1], Bin Wang[2], Andrew T. Ferenbach[1], Vladimír Farkaš[4], Ramon Hurtado-Guerrero [5,6], Javier Arroyo[3] & Daan M.F. van Aalten [1]

Fungal cell wall synthesis is achieved by a balance of glycosyltransferase, hydrolase and transglycosylase activities. Transglycosylases strengthen the cell wall by forming a rigid network of crosslinks through mechanisms that remain to be explored. Here we study the function of the *Aspergillus fumigatus* family of five Crh transglycosylases. Although *crh* genes are dispensable for cell viability, simultaneous deletion of all genes renders cells sensitive to cell wall interfering compounds. In vitro biochemical assays and localisation studies demonstrate that this family of enzymes functions redundantly as transglycosylases for both chitin-glucan and chitin-chitin cell wall crosslinks. To understand the molecular basis of this acceptor promiscuity, we solved the crystal structure of *A. fumigatus* Crh5 (*Af*Crh5) in complex with a chitooligosaccharide at the resolution of 2.8 Å, revealing an extensive elongated binding cleft for the donor (−4 to −1) substrate and a short acceptor (+1 to +2) binding site. Together with mutagenesis, the structure suggests a "hydrolysis product assisted" molecular mechanism favouring transglycosylation over hydrolysis.

[1] School of Life Sciences, University of Dundee, Dundee DD1 5EH, UK. [2] National Engineering Research Center for Non-Food Biorefinery, State Key Laboratory of Non-Food Biomass and Enzyme Technology, Guangxi Academy of Sciences, 530007 Nanning, China. [3] Departamento de Microbiología y Parasitología, Facultad de Farmacia, Universidad Complutense de Madrid, IRYCIS, 28040 Madrid, Spain. [4] Department of Glycobiology, Institute of Chemistry, Center for Glycomics, Slovak Academy of Sciences, 84538 Bratislava, Slovakia. [5] Institute of Biocomputation and Physics of Complex Systems (BIFI), University of Zaragoza, BIFI-IQFR (CSIC), 50018 Zaragoza, Spain. [6] Fundación ARAID, Av. de Ranillas, 50018 Zaragoza, Spain. Correspondence and requests for materials should be addressed to J.A. (email: jarroyo@ucm.es) or to D.A. (email: dmfvanaalten@dundee.ac.uk)

The fungal cell wall is an essential structure that maintains cell shape and protects fungi against environmental stress. Glycosyltransferases, glycoside hydrolases and transglycosylases are involved in biogenesis of the cell wall, required for growth, invading ecological niches and counteracting the host immune response[1]. Over 90% of cell wall components are polysaccharides[2], including chitin, glucan and galactomannan with varying ratios among these depending on fungal species. Chitin and β-glucan are synthesised by plasma membrane-associated synthases and extruded into the cell wall during synthesis, whereas mannoproteins are synthesised in the endoplasmic reticulum and modified in the Golgi before transport to the cell surface[3]. The final step of cell wall assembly is to generate covalent cross-links among the different cell wall components, forming a three-dimensional network mesh responsible for maintaining cell wall strength and integrity. This process takes place at the periplasmic space where polysaccharides become cross-linked by transglycosylases anchored to the plasma membrane or the cell wall[4]. Understanding the molecular mechanisms of these processes necessary for fungal cell wall formation and remodelling opens the possibility of designing novel antifungal strategies.

In *Saccharomyces cerevisiae*, the cross-links between chitin and glucan are generated by enzymes of the highly conserved Crh (Congo red hypersensitivity) family[5]. According to Carbohydrate-Active enZymes Database (CAZY), Crh enzymes belong to glycoside hydrolase family 16 (GH16). With dual chitinase and transglycosylase activities the yeast Crh enzymes generate cross-links between the reducing ends of the chitin chains and the non-reducing ends of β-1,3-glucan and β-1,6-glucan chains[4,6]. The demonstration that Crhs are required for cross-linking came from the analysis of isolated yeast cell walls, which were digested by β-1,3- or β-1,6-glucanases and subjected to size-exclusion chromatography after solubilisation by carboxymethylation to quantify the different glucan and chitin-associated fractions[5–7]. Using this approach it was shown that there is no chitin covalently bound to glucan in a strain deleted in *CRH* genes[6]. Sulphorhodamine (SR)-labelled oligosaccharides derived from β-1,3-glucan and β-1,6-glucan are incorporated as artificial acceptors for the cross-linking into the *S. cerevisiae* cell wall, showing these cross-links to be localised to bud scars and to a lesser extent to lateral cell walls[8]. The use of SR-labelled β-1,3-glucan and β-1,6-glucan oligosaccharides as acceptor sugars also allowed the detection of transglycosylase activity in vitro when soluble carboxymethyl–chitin was used as the donor polysaccharide[9]. The minimal chain length for *N*-acetyl-chitooligosaccharides as the donor was five, whereas the minimum laminarioligosaccharide acceptor chain length was two glucose units[9]. In contrast to the strict donor specificity (only CM–chitin or chitooligosaccharides are accepted as donors), SR-labelled β-1,3- and β-1,6- glucooligosaccharides and SR-labelled *N*-acetyl-chitooligosaccharides both acted as acceptors[9]. The most efficient acceptor was CH4-SR (chitotetraose-SR) followed by L4-SR (laminaritetraose-SR), whereas the transfer to β-1,6-linked glucooligosaccharides (P4-SR) was less favoured[9]. These preferences correlate well with the incorporation of the corresponding labelled oligosaccharides in culture, suggesting that Crh enzymes could act not only as heterotransglycosylases attaching chitin chains to β-1,3-glucan and β-1,6-glucan, but also as homotransglycosylases to form chitin to chitin cross-links[4,9].

The Crh family is a conserved group of enzymes unique to fungi. In *S. cerevisiae*, there are three members of this family, Crh1, Crh2 and Crr1. Crh1 and Crh2 are functional during vegetative growth[10] whereas Crr1 is involved in spore cell wall synthesis[11]. Incorporation of SR-labelled oligosaccharides into the cell wall is completely abolished in a double *crh1Δcrh2Δ*

mutant[9,12]. Deletion of either *CRH1* or *CRH2* resulted in a defective cell wall indicated by hypersensitivity to Congo Red and Calcofluor white (CFW) while the *crh1Δcrh2Δ* double mutant exacerbated the phenotype[10]. Together with the septin ring, yeast chitin–glucan crosslinks mediated by Crh proteins play a role in the control of morphogenesis by preventing cell wall growth at the mother-bud neck region through the cell cycle[13,14]. In *Candida albicans*, three members (Utr2, Crh11 and Crh12) belong to the Crh family. Similar to *S. cerevisiae*, single, double and triple deletions of these genes led to hypersensitivity to cell wall disrupting agents[15]. Furthermore, the triple mutant was avirulent in a mouse infection model although the colonisation was unaffected[15]. Multiple sequence alignment also showed five Crh orthologues in *Neurospora crassa* and five in *Aspergillus fumigatus*[4]. The redundancy and conservation of the catalytic residues of the Crh family in these fungi support transglycosylation by these proteins as a common fungal mechanism to cross-link cell wall components and suggest a similar role for fungal Crh proteins in cell wall remodelling.

Although Crh enzymes share sequence and structure similarities with other enzymes of the GH16 family[4], the structure of these enzymes, the molecular basis of sugar substrate recognition and the catalytic mechanism are undefined. Recently the structure of *S. cerevisiae* Crh1 was modelled and a catalytic mechanism proposed[12]. The transglycosylase activity of these proteins would involve the cleavage of β-1,4-glycosydic linkages of the chitin and subsequent attachment of the fragment from the donor molecule with the newly formed reducing end onto the O4 hydroxyl group of the acceptor molecule by a β-1,4-glycosydic bond[9]. However, a comprehensive understanding of the catalytic mechanism and particularly the acceptor substrate promiscuity of these enzymes requires an experimentally determined structure of a Crh enzyme in complex with its substrates.

Here, we structurally and functionally characterise the Crh family in the filamentous fungus *A. fumigatus*, an opportunistic fungal pathogen causing invasive infections in immunodeficient individuals. Using soluble chitin derivatives as oligoglycosyl donor and oligosaccharides derived from chitin and β-1,3-glucan as acceptors, *A. fumigatus* Crh enzymes act as transglycosylases for formation of chitin–chitin and chitin–glucan crosslinks both in vitro and in vivo. The structure complexed with sugars reveal a typical β-jelly roll fold with the donor and acceptor sugar-binding sites located at the concave face. Site-directed mutagenesis in combination with superposition allow us to understand not only the catalytic mechanism of transglycosylation but also uncover the molecular basis of the observed acceptor promiscuity of this family.

## Results

### *A. fumigatus* possesses a family of putative cell wall cross-linking enzymes.
The Crh transglycosylases have been extensively analysed in yeast, but the extent of this family and its function remains underexplored in filamentous fungi. Using *S. cerevisiae* Crh1 (Uniprot: P53301) and Crh2 (Uniprot: P32623) as sequences for BLAST searches[16] in the *A. fumigatus* genome revealed five potential orthologues: *Af*Crh1 (Q4WD22), *Af*Crh2 (Crf2, Q4WI46), *Af*Crh3 (Q4WXE6), *Af*Crh4 (Q4WMW2) and *Af*Crh5 (Crf1, Q8J0P4). All five gene products have signal peptides at the *N*-terminus while *Af*Crh1, *Af*Crh2 and *Af*Crh5 also have GPI anchor sites at the *C*-terminus, as predicted by the SignalP 4.1[17] and PredGPI[18] servers, respectively. Apart from *Af*Crh4 and *Af*Crh5 other *Af*Crh enzymes have multiple glycosylation sites according to NetNGlyc 1.0 Server prediction http://www.cbs.dtu.dk/services/NetNGlyc. Sequence alignment of the *A. fumigatus* Crh enzymes with the *S. cerevisiae* and *C. albicans*

Crh enzymes revealed two motifs, the DEXDXE motif and the GTIXWXGG motif, that are highly conserved in all Crh proteins (Supplementary Fig. 1). Similar to *Sc*Crh2 in *S. cerevisiae* and Utr2 orthologues in *C. albicans*, *Af*Crh2 contains a family 18 chitin-binding module (CBM18) after the *N*-terminal signal peptide, that has a functional role in tuning Crh activity in yeast[12]. Hence, *A. fumigatus* possesses a family of putative cell wall cross-linking enzymes.

### Crh family is dispensable for *A. fumigatus* viability in vitro.

We next aimed to investigate the role of these enzymes in *A. fumigatus* using a genetic approach. In order to make single and multiple knockouts for all five *crh* genes, a single but recyclable *pyrG* marker was chosen for mutant selection. This marker was flanked by two *neo* fragments, which not only confer kanamycin resistance during construct generation but also account for intra-chromosomal homologous recombination when the *pyrG* marker needs to be excised for reuse (Supplementary Fig. 2a). Through iterative protoplast transformation and mutant selection cycles all single *crh* mutants (Δ*crh1*, Δ*crh2*, Δ*crh3*, Δ*crh4* and Δ*crh5*), double (Δ*crh1*Δ*crh2*), triple (Δ*crh1*Δ*crh2*Δ*crh3*), quadruple (Δ*crh1*Δ*crh2*Δ*crh3*Δ*crh5*) and quintuple mutant (Δ*crh1*Δ*crh2*Δ*crh3*Δ*crh4*Δ*crh5*) strains were obtained and confirmed (Supplementary Fig. 2b and c). All the mutants were analysed in terms of growth rate and sensitivity to the cell wall disrupting agents such as Congo red[19]. Unexpectedly, unlike the *crh* mutants in *S. cerevisiae*[5,10] and *C. albicans*[15], all *crh* mutants in *A. fumigatus* (including the quintuple mutant) displayed only minor sensitivity to high concentrations of CR (Supplementary Fig. 3). No defects in growth rate, germination or sporulation were detected for the mutants compared to the parental strain in

our experimental conditions. Thus, it appears that the Crh family is dispensable for *A. fumigatus* viability in vitro.

### The redundant Crh family is required for creating cross-links at the septa and cell wall.

Previous studies have analysed the cross-linking between chitin and β-1,3-glucan in living yeast cells by using SR-linked oligosaccharides as artificial chitin acceptors[8,9]. To investigate this cross-linking processes in *A. fumigatus*, WT and mutant strains were grown in the presence of SR, SR-linked laminaripentaose (L5-SR), SR-labelled chitopentaose (CH5-SR) or FITC-labelled chitohexose (NAG6-FITC) and their incorporation was analysed by fluorescence microscopy. As shown in Supplementary Fig. 4, free SR did not enter cells whereas L5-SR was incorporated into the WT lateral cell walls and accumulated at the septa, as revealed by simultaneous CFW staining (Fig. 1). The growth of WT cells in the presence of CH5-SR resulted in a higher incorporation and in a similar localisation pattern to that of the L5-SR signal (Fig. 1, right panel), suggesting that, as previously shown in yeast, the Crh family in *A. fumigatus* not only transglycosylate chitin to glucan but also chitin to chitin. The NAG6-FITC labelling exhibited the same incorporation pattern as the L5-SR and CH5-SR (Supplementary Fig. 5). Interestingly, the fluorescence was completely abolished in the quintuple mutant (Δ*crh1*Δ*crh2*Δ*crh3*Δ*crh4*Δ*crh5*) whereas the quadruple mutant (Δ*crh1*Δ*crh2*Δ*crh3*Δ*crh5*) exhibited fluorescent signal similar to the WT strain (Fig. 1 and Supplementary Fig. 5). Moreover, in a revertant strain where the *crh4* or *crh3* gene was reintroduced into the quintuple mutant, the L5-SR, CH5-SR and NAG6-FITC signals were restored (Fig. 1 and Supplementary Fig. 5). Thus, *A. fumigatus* Crh enzymes show a redundant activity required for polysaccharide cross-linking at the septa

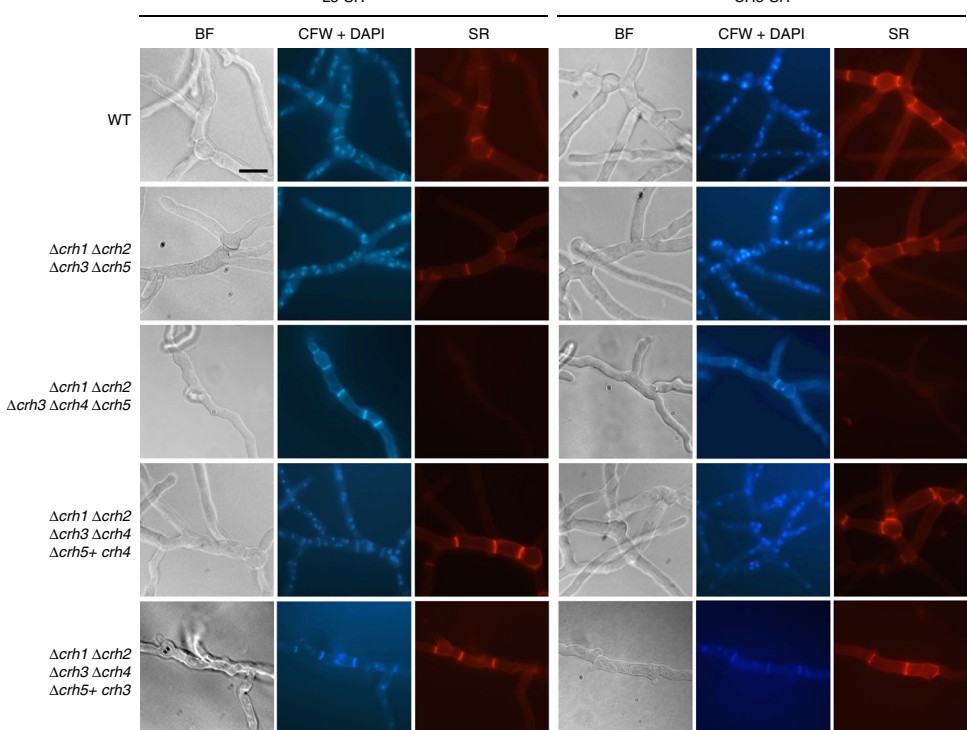

**Fig. 1** Incorporation of different SR-oligosaccharides is blocked in the quintuple *crh* mutant. $10^5$ conidia of WT and the indicated strains were incubated with 3.75 µM of SR-labelled laminaripentaose (L5-SR) or SR-labelled penta-acetyl chitopentaose (CH5-SR) for 16 h at 37 °C. Cells were fixed and stained with 10 µg ml$^{-1}$ DAPI and 10 µg ml$^{-1}$ CFW before being analysed by fluorescence microscopy. Panels (from left to right) were the same cells under bright field (BF), UV channel (CFW+DAPI) and the rhodamine fluorescence channel (SR), as indicated. All images were taken at the same exposure. Scale bar, 10 µm

and cell wall as deduced from the lack of cross-linking in the quintuple mutant and the normal incorporation of oligo-saccharides both in the quadruple mutant and in the quintuple revertant strain expressing *crh4* or *crh3*. Therefore, in the absence of any of the individual Crh proteins, the other family members are able to transglycosylate the corresponding polysaccharide.

**AfCrh5 is a transglycosylase with acceptor substrate promiscuity**. To allow evaluation of Crh activity in vitro we next sought to develop a recombinant expression system. Previously, *Af*Crh5 (residues 1–370, also called Crf1) was expressed in *Pichia pastoris* and identified as a fungal antigen[20]. Considering that *Af*Crh5 is predicted to not contain *N*-glycosylation sites, we evaluated its overexpression in *Escherichia coli*. With the aim of

optimising expression levels, a range of expression constructs with different boundaries were tried. Finally, a truncated form of *Af*Crh5 (residues 22–275, excluding the signal peptide, the Ser/Thr-rich region and GPI-anchor sequences) was expressed as a fusion protein featuring an *N*-terminal PreScission cleavable GST-tag followed by a His-tag. After purification by GST beads, PreScission protease cleavage and gel filtration chromatography, pure *Af*Crh5 with a non-cleavable His-tag was obtained at a yield of 3.4 mg L$^{-1}$.

Using a recently developed sensitive fluorescence assay[9], the chitin to glucan transglycosylation activity of *Af*Crh5 was measured. As shown in Fig. 2a, when carboxymethyl–chitin (CM–chitin, 0.1%) was used as the donor sugar and L5-SR as the acceptor sugar, the optimum pH for transglycosylation was between 4.3 and 4.9 (Fig. 2a). pH 4.9 was chosen for all

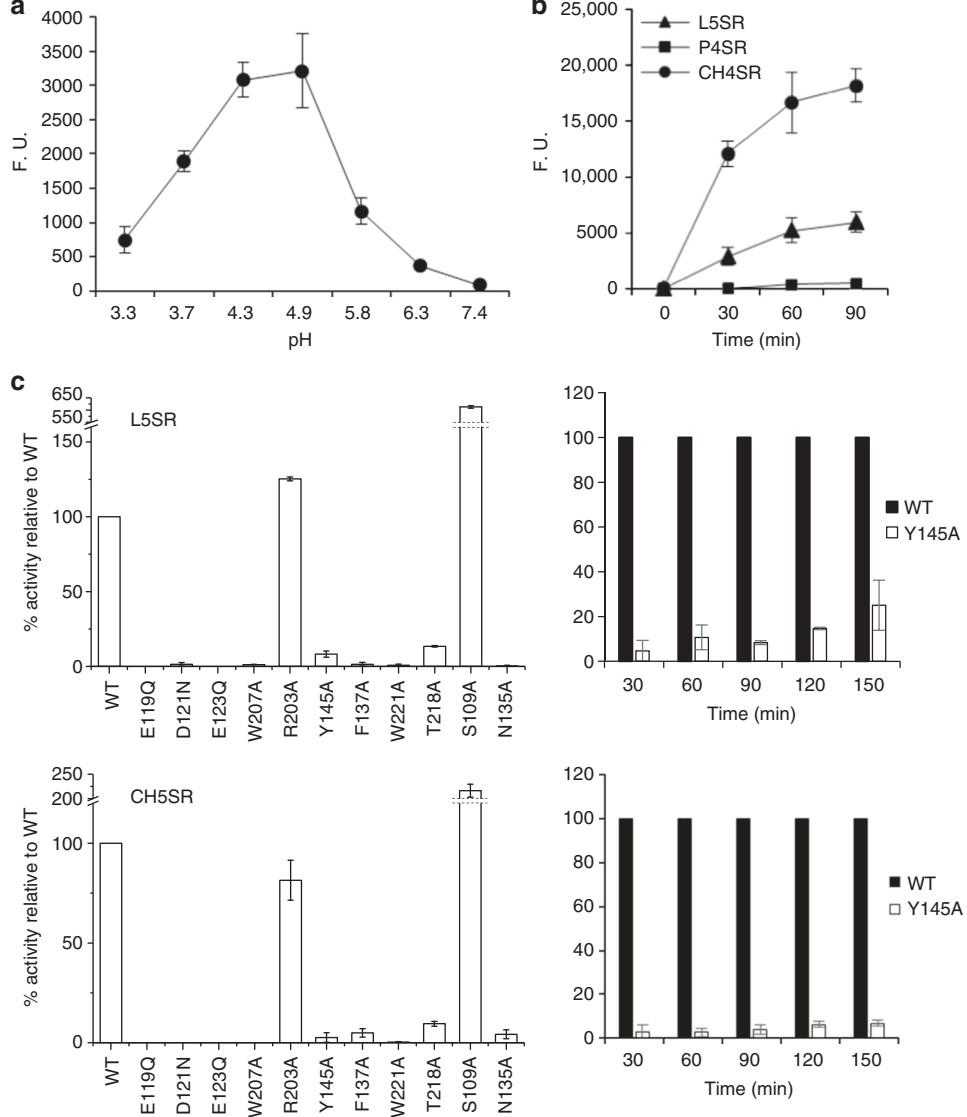

**Fig. 2** In vitro transglycosylase activity of *Af*Crh5 and mutant proteins. **a** Determination of optimum pH for *Af*Crh5 transglycosylase activity measured after 60 min of reaction using 18 µM SR-labelled laminaripentaose (L5-SR) as acceptor and CM-chitin (0.1%) as donor in defined 50 mM citrate buffer. F.U. represents arbitrary fluorescence units determined for each time point after substracting the value at the time of adding the enzyme (time 0). **b** Comparison of transglycosylation rates catalysed by *Af*Crh5 using equimolar concentrations of the respective SR-labelled oligosaccharides as acceptors and CM–chitin as a donor. The reactions were carried out under standard conditions described in Methods at the indicated times using 4.9 as optimum pH. **c** Transglycosylase activity of *Af*Crh5 mutant proteins relative to the WT activity (100%) using L5-SR or CH5-SR as acceptors was measured as in **b** after 90 min of reaction. For the Y145A mutant a time course enzymatic activity experiment including longer incubation time is also included. Data represent the average and standard deviation of at least three independent experiments. Source data are provided as a Source Data file

**Table 1 Kinetic parameters of *Af*Crh5 with L5-SR and CH5-SR as the respective acceptors**

| Enzyme | Acceptor | $K_m$ (μM) | $V_{max}$ (pmol s$^{-1}$ ml$^{-1}$) | $k_{cat}$ (s$^{-1}$) (×10$^{-3}$) | $k_{cat}/K_m$ (s$^{-1}$ μM$^{-1}$) (×10$^{-3}$) |
|---|---|---|---|---|---|
| *Af*Crh5 | L5-SR | 12 ± 3 | 0.17 ± 0.04 | 0.06 ± 0.01 | 0.005 |
| | CH5-SR | 3.5 ± 0.4 | 1.0 ± 0.2 | 0.31 ± 0.07 | 0.09 |

The average values ± S.D. from three independent measurements are given

subsequent experiments. When using different SR-labelled sugars as acceptors, including laminarin-derived oligosaccharides, pustulan-derived oligosaccharides and chitooligosaccharides, *Af*Crh5 showed higher transglycosylation efficiency towards SR-labelled chitopentaose (CH5-SR) than L5-SR, whereas SR-labelled pustulotetraose (P5-SR) was barely linked to the CM-chitin (Fig. 2b). Michaelis constants showed that from the turnover rate ($k_{cat}$) and $k_{cat}/K_m$ ratios of the enzyme for both substrates (CH5-SR and L5-SR) (Table 1), *Af*Crh5 appears to be more catalytically efficient towards chitooligosaccharides than laminarioligosaccharides. Thus, *Af*Crh5 is a transglycosylase with acceptor substrate promiscuity, including crosslinking chitin to chitin.

**_Af_Crh5 establishes extensive substrate interactions in the −2 to +2 subsites.** Despite the functional characterisation of Crh proteins as enzymes necessary for fungal cell wall assembly and morphogenesis in *S. cerevisiae*[13] and virulence in *C. albicans*[15], it is not understood how this class of enzymes achieves the observed substrate promiscuity and favours transglycosylation over hydrolysis. To address these questions, we determined the apo crystal structure of *Af*Crh5 and its crystal structure in complex with chitooligosaccharides. Apo crystals were obtained using ammonium sulphate as a precipitant and diffracted to 2.25 Å. Various chitooligosaccharides and gluco-oligosaccharides were soaked with the apo crystals. The best of these was a chitin tetrasaccharide (NAG4) soak displaying diffraction to 2.8 Å. The apo and the complexed structure were solved by molecular replacement and refined to 0.23 and 0.24 $R/R_{free}$ values, respectively (Table 2). Both crystals belong to the same orthorhombic space group with two independent molecules in the asymmetric unit (AU). No conformational changes were observed between the apo and the complexed *Af*Crh5 structures or between the AU molecules with overall root-mean-square deviation (RMSD) values of 0.20 and 0.25 Å for 213 Cα atoms, respectively. *Af*Crh5 adopts the GH16 family β-jelly-roll fold composed of two stacked seven-stranded β-sheets with a convex and a concave face (Fig. 3a). Initial electron density defined the positions of NAG4 and chitobiose (NAG2), which were located in the donor (−4 to −1) and acceptor (+1 to +2)-binding sites on the concave face, respectively (Fig. 3a). Interpreting this structure in the context of the sequence alignment of the *A. fumigatus* Crh family (Supplementary Fig. 1) reveals that most residues located in the catalytic region and sugar-binding pocket are conserved (Fig. 3b). *Af*Crh5 shares the general EXDXE active site motif present in most GH16 family enzymes in which the first glutamate, the middle aspartate and the last glutamate act as the catalytic nucleophile, the auxiliary residue, and the general acid/base, respectively[21–24]. In *Af*Crh5, Glu119 and Glu123 are predicted to act as the nucleophile and the general acid/base (Fig. 3c), respectively. Four aromatic residues, Phe137, Trp207, Trp221 and Tyr145, together with the catalytic residues delineate the sugar-binding groove (Fig. 3c). The −1 O6 hydroxyl interacts with Glu123, Ser109 and Trp207 side chains, and the −1 O1 hydroxyl interacts with Asp121 (Fig. 3c). The sugar moiety in the −2 subsite establishes two types of interactions: a CH−π interaction between the sugar moiety and Trp207, and hydrogen

**Table 2 Summary of data collection and structure refinement statistics**

| Data collection | Apo *Af*Crh5 | *Af*Crh5 in complex with NAG4 |
|---|---|---|
| Space group | P 2$_1$ 2$_1$ 2$_1$ | P 2$_1$ 2$_1$ 2$_1$ |
| *a, b, c* (Å) | 55.4 74.2 116.8 | 55.4 74.2 116.8 |
| Resolution range (Å) | 62.62–2.25 (2.33–2.25) | 55.39–2.80 (2.95–2.80) |
| Unique reflections | 23361 (2128) | 12525 (1226) |
| Mean$I/\sigma(I)$ | 6 (1.9) | 10.4 (3.7) |
| Multiplicity | 4.1 (4.2) | 4.0 (4.0) |
| Completeness (%) | 99.4 (99.7) | 99.4 (99.5) |
| $R_{sym}$[a] | 0.17 (0.76) | 0.11 (0.36) |
| $R_{pim}$[b] | 0.10 (0.47) | 0.079 (0.27) |
| *Refinement* | | |
| $R_{work}/R_{free}$[b] | 0.19/0.23 | 0.19/0.24 |
| Number of non-hydrogen atoms | 3936 | 3892 |
| Macromolecules | 3664 | 3674 |
| Ligands | 42 | 172 |
| Solvent | 230 | 46 |
| Protein residues | 483 | 484 |
| *B-factor (Å$^2$)* | | |
| Macromolecules | 28.4 | 28.9 |
| Ligands | 51.2 | 51.3 |
| Solvent | 32.6 | 14.2 |
| *R.m.s. deviations* | | |
| RMS(bonds) | 0.013 | 0.013 |
| RMS(angles) | 1.95 | 2.12 |
| PDB code | 6IBU | 6IBW |

Values in parenthesis are for the highest resolution shell. All measured data were included in structure refinement
[a]$R_{sym} = \Sigma_h \Sigma_i \ |I_{hi} - \langle I_h \rangle| / \ \Sigma_h \Sigma_i I_{hi}|$, where $I_{hi}$ is the intensity of the *i*th measurement of the same reflection and $\langle I_h \rangle$ is the mean observed intensity for that reflection
[b]$R_{pim} = \Sigma_h [1/(N−1)] \ 1/2 \ \Sigma_i |\langle I_{hi} \rangle − [I_h]|/\Sigma_h \Sigma_i I_{hi}|$, where $\Sigma_i I_{hi}$ is the *i*th measurement of reflection h, $\langle I_h \rangle$ is the mean observed intensity of all measurements and N is the redundancy for the h reflection
[c]$R_{work} = \Sigma_h ||F_{obs}| − |F_{calc}||/\Sigma_h |F_{obs}|$, where $F_{calc}$ and $F_{obs}$ are the observed and calculated structure factors for the reflection h
[d]$R_{free}$ is equivalent to $R_{work}$ calculated with a reserved 5% of the reflections

bonds between Arg203 and the carbonyl group and between water molecules and the O3 hydroxyl/amide group (Fig. 3c). The sugar moieties located at −3 and −4 subsites do not establish any direct interactions with the enzyme and only the −3 carbonyl group is engaged to a water molecule by a hydrogen bond interaction. To further analyse this observation, we calculated the omit maps for the ligand and further estimated relative occupancy in the −4 and −3 subsites to be 60% and 80%, respectively (Supplementary Fig. 6 and Table 2). The +1 sugar moiety is tethered by Trp221, Asn135 and Glu123. There are CH−π interactions between the sugar moiety and Trp221, and hydrogen bonds between the O3 hydroxyl with Asn135, and the O4 hydroxyl with Glu123 (Fig. 3c). Finally, the +2 sugar moiety establishes a CH−π interaction with Tyr145 and a hydrogen bond between the O3 hydroxyl and Thr218 (Fig. 3c). Therefore, Crh enzymes possess a well-conserved donor and acceptor substrate binding site, establishing extensive substrate interactions in the −2 to +2 subsites.

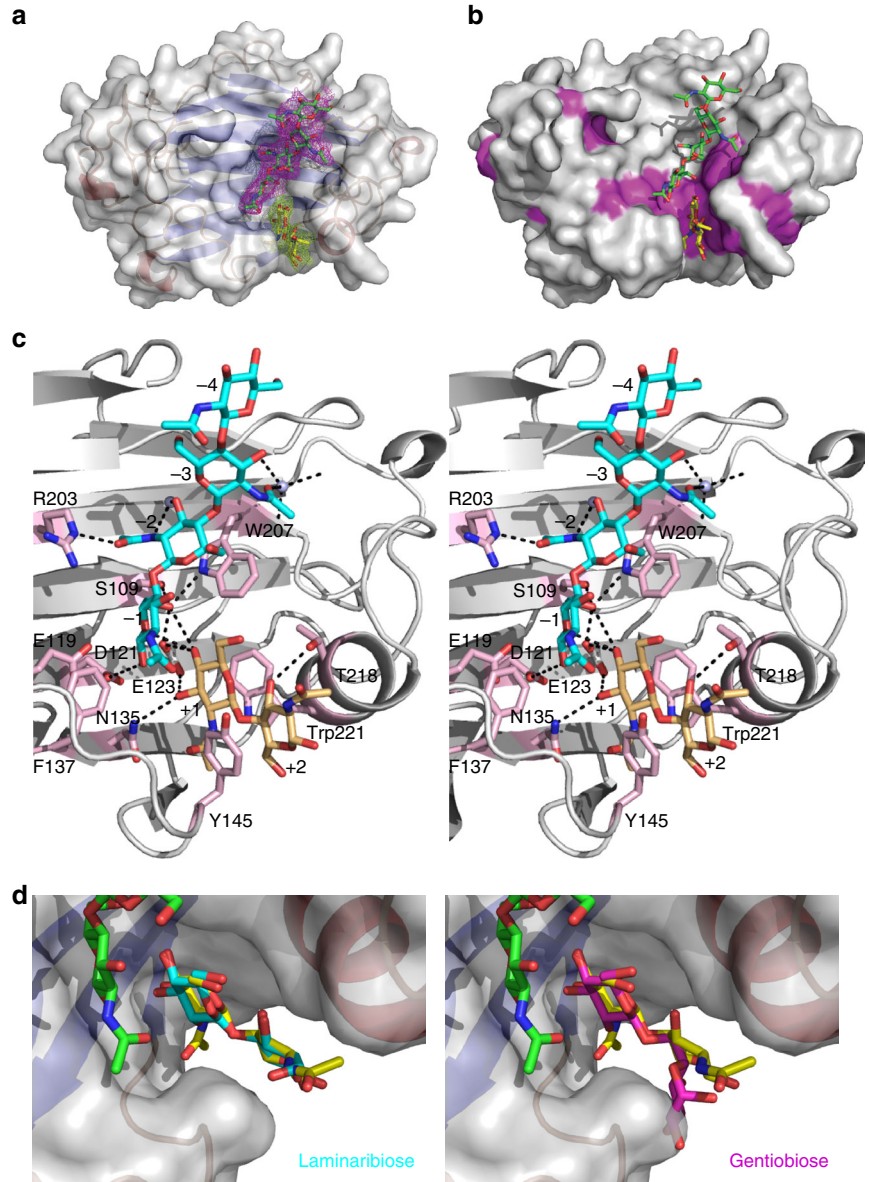

**Fig. 3** Crystal structure of *Af*Crh5 complexed with NAG4 and superposition of laminaribiose and gentiobiose structures on to NAG2-binding site. **a** Overall fold of *Af*Crh5 (residues 22–275) in complex with NAG4. Protein surface is coloured in grey. Secondary structure elements are coloured red (helices) and blue (strands). Donor NAG4 is shown as sticks with green $C_\alpha$ while acceptor NAG2 is shown in yellow $C_\alpha$ sticks. The unbiased $|F_o|-|F_c|$ map (2.25$\sigma$) is shown as magenta mesh for NAG4 and yellow mesh for NAG2. **b** Surface representation of conserved residues in *A. fumigatus* Crh family. *Af*Crh5 shown in grey surface, residues conserved in other Crh enzymes are coloured in magenta. Donor NAG4 is shown as sticks with green $C_\alpha$ while acceptor NAG2 is shown in yellow $C_\alpha$ sticks. **c** Stereoscopic view of the active site of *Af*Crh5 in complex with donor NAG4 and acceptor NAG2. Donor NAG4 is shown as sticks with cyan $C_\alpha$ while acceptor NAG2 is shown as yellow orange $C_\alpha$ sticks. Hydrogen bonds are shown in black dashed lines between NAG4, NAG2 and their interacting residues, whose side chains are shown as sticks with light pink $C_\alpha$ atoms. **d** Surface representation of an overlay of the experimentally determined NAG2 acceptor position with a modelled laminaribiose and gentiobiose. The experimentally determined NAG2 acceptor position is shown in yellow $C_\alpha$ sticks whereas superposed laminaribiose is shown in cyan $C_\alpha$ sticks (left) and superposed gentiobiose is shown in magenta $C_\alpha$ sticks (right)

***Af*Crh5 achieves substrate promiscuity through a conserved tyrosine.** Previous work with the yeast Crh enzymes revealed that the donor specificity of these enzymes is restricted to soluble chitin derivatives. However, the enzymes exhibited considerable promiscuity for the acceptor substrates, tolerating oligosaccharides derived from either β-1,3-linked or β-1,6-linked gluco-oligosaccharides and even chitooligosaccharides albeit with different catalytic efficiencies[4,9]. The most efficient acceptors in vitro were chitin followed by laminarin derivatives, suggesting that in addition to the heterotransglycosylation of chitin onto β-1,3-linked and β-1,6-glucan, these enzymes could also act as homo-

transglycosylases joining nascent chains of chitin[4,9]. In *A. fumi-gatus*, *Af*Crh5 drives crosslinking of chitin to β-1,3-glucan or chitin (Figs. 1 and 2) while no crosslinking of chitin to β-1,6-glucan was observed, in agreement with the lack of β-1,6-glucan in this organism. *Af*Crh5, like *Sc*Crh1 and *Sc*Crh2, transfers chitin residues onto chitin more efficiently than chitin onto β-1,3-glucan (Table 1). To explore the molecular basis of the promiscuity of these enzymes, we soaked combinations of chitooligosaccharides with β-1,3-glucan or β-1,6-glucan oligosaccharides into our *Af*Crh5 crystals, but we were unable to displace the chitobiose from the acceptor site. As an alternative, we performed

superposition of laminaribiose and gentiobiose from PDB entries 1OD3[25] and 4ZO7[26] in the acceptor binding site and compared it to the NAG2 conformation observed in our complex (Fig. 3d). This superposition approach appears to suggest that laminaribiose can be accommodated (Fig. 3d) while gentiobiose cannot, due to a different arrangement of the +2 glucose as a result of the β-1,6 linkage (Fig. 3d). Both the NAG2 complex and the superposed laminaribiose shared the same key CH–π interaction between the +2 sugar moiety and Tyr145, whereas the +2 sugar moiety of gentiobiose presented steric hindrance with Tyr145 and lost a hydrogen bond with T218 (Fig. 3d). This explains why AfCrh5 prefers chitin and β-1,3-glucan as acceptor substrates in contrast to β-1,6-glucan. Mutagenesis of Tyr145 to Ala resulted in a reduction of the transglycosylase activity for both CH5-SR and L5-SR as acceptors (Fig. 2c). Under conditions with extended reaction times, the Y145A mutant maintained 24% activity using L5-SR as acceptor, whereas it had only 8% activity when CH5-SR acted as acceptor (Fig. 2c). Tyr145 is highly conserved along the Crh family of enzymes including those containing CBM domains (an exception is found in AfCrh3 that contains a Phe residue instead) (Supplementary Fig. 1). Thus, AfCrh5 achieves substrate promiscuity through a conserved tyrosine.

**AfCrh5 disfavours hydrolysis by protection of the acceptor-binding site.** One of the key challenges of transglycosylases is to protect the intermediate against premature hydrolysis. We next explored how AfCrh5 achieves this. The 5.6 Å distance between the nucleophile (Glu119) and the general acid/base (Glu123) in AfCrh5 suggests that the active site structure is compatible with a retaining mechanism involving a covalent enzyme–donor intermediate[27]. This is further evidenced by the distances of 3.5 Å between −1 O1 hydroxyl and the nucleophile Glu119, and 2.8 Å between +1 O4 hydroxyl and the acid/base Glu123 (Fig. 3c). This mechanism is supported by site-directed mutagenesis of the nucleophile Glu119 and acid/base Glu123 to Gln, resulting in abrogation of AfCrh5 transglycosylase activity with both L5-SR and CH5-SR (Fig. 2c) as acceptor substrates. Mutagenesis of Asp121 to Asn also rendered AfCrh5 enzymatically inactive, in agreement with the importance of the EXDXE motif in GH16 enzymes[22,24]. The role of the two aromatic residues Trp207 and Trp221, which establish CH–π interactions with −1 and +1 sugar moieties, was also evaluated by site-directed mutagenesis. Both residues were mutated to Ala, and the resulting mutants exhibited no enzymatic activity, confirming their essential role in substrate recognition (Fig. 2c). Mutagenesis of acceptor sugar interacting residues, such as Thr218 and Tyr145 to Ala resulted in a severe reduction of catalytic activity (Fig. 2c), suggesting they are important for acceptor sugar binding. Mutation of Arg203 to Ala did not lead to significant changes in the activity, suggesting that the −2 sugar moiety is sufficiently tethered by a CH–π interaction with Trp207 and hydrogen bonding to water molecules. Mutation of Asn135, whose amide group interacts with +1 O3 hydroxyl group, led to reduction of activity towards L5-SR or CH5-SR, in agreement with its structural role in coordinating the +1 sugar. Surprisingly, mutation of Ser109 to Ala made AfCrh5 four-fold more active than the wild type enzyme (Fig. 2c). It is possible that this mutation may position the −1 sugar moiety closer to Trp207 or stabilise the basic character of Glu123 during the catalytic cycle. Interestingly, although Phe137 did not interact directly with sugar moieties, its mutation eliminated transglycosylase activity (Fig. 2c). Phe137 provides a hydrophobic environment for Glu119 and may affect its $pK_a$.

To study the effects of these mutations on hydrolysis we determined the chitinase activity for the wild type and mutant enzymes. We used a <u>f</u>luorophore-<u>a</u>ssisted <u>c</u>arbohydrate

electrophoresis (FACE) assay to detect chitinase activity using NAG2 to NAG5 as the substrates followed by 8-aminonaphthalene-1,3,6-trisulphonic acid (ANTS) labelling. The data showed that AfCrh5 has poor chitinase activity towards NAG5 and no activity for NAG2, NAG3 and NAG4 (Supplementary Fig. 7a). In addition, the effects of the AfCrh5 mutations on chitinase activity followed the same trends as the effects on transglycosylase activity (Fig. 2), implying that the chitinase and transglycosylase activities are coupled (Supplementary Fig. 7b). Furthermore, FACE analysis revealed the presence of transglycosylated products (faint bands of higher oligomers) when AfCrh5 was incubated with NAG5 as a substrate alone (Supplementary Fig. 7c) or in combination with laminarioligosaccharides (G4 or G5) (Supplementary Fig. 7d), in agreement with the transglycosylase activity measured in the fluorescence assay. As NAG5 is the minimum donor required for transglycosylation, it is likely the electron density in the acceptor site of the complex structure represents two β-1,4-linked N-acetylglucosamine sugars, with the additional two occupying the +3 and +4 subsites being disordered. Thus reactions are unlikely to have taken place in the crystal.

In the proposed double displacement retaining reaction mechanism, a covalent glycosyl-enzyme intermediate is formed between the −1 sugar moiety anomeric carbon and Glu119, which is subsequently cleaved by the incoming acceptor substrate leading to the formation of a new β-1,4-glycosidic linkage[12]. To avoid hydrolysis, the enzyme must employ specific strategies to exclude the surrounding water molecules from the covalent glycosyl-enzyme intermediate favouring transglycosylation. The crystal structure together with the kinetics studies on the mutants implies that initially the substrates NAG5 and chitin are hydrolysed releasing shorter and long NAG products, respectively. As suggested by the crystal structure, these hydrolysis products may still bind the +1/+2 acceptor subsites, thus excluding potential nearby water molecules that would lead to premature hydrolysis of the covalent glycosyl–enzyme intermediate. It is possible that these products are displaced from the active site simultaneously with an incoming polysaccharide acceptor, exploiting the exposed aromatic residues in the binding site. This mechanism would be even more favoured in the cell wall space due to the limited access of the enzyme bulk solvent in this compartment. Overall, our results allow us to propose that these enzymes are transglycosylases that circumvent hydrolysis by a synergistic approach of combining their location in a hydrophobic environment with a mechanistic strategy that deals with the hydrolysis products protecting the covalent glycosyl-enzyme intermediate and in turn driving transglycosylation.

## Discussion

Fungal cell wall biogenesis has long been considered a possible drug target due to its essential role in fungal biology and absence of similar structures in mammalian cells[28,29]. In the past decades, studies have focused on the molecular characterisation, organisation and biosynthesis of cell wall[30,31]. However, there have been few studies aimed at understanding cross-linking enzymes, limited to the Crh enzymes in S. cerevisiae and some other enzymes such as the large family of transglycosylases for elongating β-1,3-glucan (e.g. Gas family in S. cerevisiae[32], Gel family in A. fumigatus[33–35] and Phr family in C. albicans[36]), transglycosylases for branching β-1,3-glucan (e.g. Bgt family and Gel4 in A. fumigatus[32,37]), and transglycosylases for covalent linking of galactomannan to the β-1,3-glucan–chitin cell wall core (e.g. Dfg family in A. fumigatus and N. crassa[38]). Here we have combined genetics, biochemistry and structural biology approaches to give a comprehensive understanding of the cell wall crosslinking Crh enzymes in the filamentous fungus A. fumigatus.

Unlike Crh mutants from *S. cerevisiae* and *C. albicans* in which single and multiple knockout mutants displayed hypersensitivity to Congo red[5], morphological aberrations at the mother-bud neck[14] and avirulence in mouse models of infection[15], all Crh mutants in *A. fumigatus* displayed only slight sensitivity to high concentrations of CR (Supplementary Fig. 3) and no difference in terms of growth rate, germination, sporulation when compared to the parental strain. These data called into question whether the Crh enzymes perform cell wall cross-links in *A. fumigatus* or suggested that other, as yet unidentified enzymes, redundantly possess activities to crosslink polymers of the cell wall. Interestingly, using fluorescently labelled acceptor sugars in culture, we showed that the *A. fumigatus* Crh enzymes were active as cell wall transglycosylases. The cross-linking occurred at the cell wall and was particularly strong at the septa but without affecting septa formation (Fig. 1 and Supplementary Figs. 4 and 5). The signal for cross-linking of both chitin onto glucan and chitin onto chitin was totally abolished in the quintuple mutant, suggesting that there were no other enzymes responsible for this type of cross-linking apart from the Crh family. Despite the abundant cross-linking in the septum, there were no morphogenesis defects in the quintuple mutant. However, as shown in yeast the morphogenic phenotype was only observed in a $\Delta cla4\Delta crh1\Delta crh2$ strain[13], thus, future work focusing on the function of other, possibly redundant, septum or cell wall proteins in *A. fumigatus* is necessary. Nevertheless, our research here demonstrates that the function of Crh family in filamentous fungus is less important for viability than it is in yeast. However, it is quite possible that the Crh family may be important for pathogenesis during infection in a host. These enzymes are mostly located at the surface of the cell wall, the first point of contact with immune cells of the host—indeed they induce the production of specific antibodies[20], trigger CD4$^+$ T$_H$1 cells[39] and directly bind to cytokines, such as IL-17A to facilitate survival, adaptation and virulence[40].The physiological role of *A. fumigatus* Crh family requires further investigation.

Recombinant *Af*Crh5 displayed predominantly in vitro transglycosylase activity and to a lesser extent chitinase activity towards NAG5 (Fig. 2 and Supplementary Fig. 7a and d). Only longer chitooligosaccharides (≥NAG5) act as the donor for chitinase activity (Supplementary Fig. 7a), and this step necessarily precedes transglycosylation to form the covalent enzyme–donor intermediate. Thus, both the in vitro transglycosylation assay and in culture labelling revealed that the Crh activity is conserved among yeast and *Aspergillus*. Aiming to understand the catalytic mechanism of this family of enzymes, we solved the crystal structure of *Af*Crh5 in complex with NAG4 and NAG2 in the donor and acceptor binding sites, respectively. As NAG5 is the minimum donor required for transglycosylation, reactions are unlikely to have taken place in the crystal. The density in the acceptor site of the complex structure actually represents two β-1,4-linked *N*-acetylglucosamine sugars, with the additional two occupying the +3 and +4 subsites being disordered. This is the first structure of this family of enzymes and allowed us to dissect the role of residues interacting with the sugar moieties by site-directed mutagenesis experiments and subsequent kinetics studies. Mutations of Tyr145, Trp221, Phe137 and Thr218 to Ala affected not only the transglycosylase activity but also the chitinase activity (Fig. 2c and Supplementary Fig. 7b). In addition, based on the crystal structure and the mutations we propose that the critical residues Tyr145 and Trp221 form a "slide" allowing the chitinase cleaved product shifting out without fully dissociating from the active site while the new acceptor approaches the binding site. This seamless deposition would protect the glycosyl-enzyme intermediate from water molecules, favouring transglycosylation over hydrolysis. This would be further enhanced taking into account that the nature of the cell wall is highly hydrophobic and abundant in chitin and β-1,3-glucan.

Although initially these enzymes were identified as chitin–glucan crosslinking enzymes[10], recent progress[9] and our own experimental data have shown that Crh enzymes are capable of conducting both chitin–glucan and chitin–chitin transglycosylation in culture and in vitro (Figs. 1, 2 and Supplementary Fig. 7c, d). However, *Af*Crh5 could not perform chitin to β-1,6-glucan crosslinks (Fig. 2b), in line with the absence of β-1,6-glucan in *A. fumigatus*. Superposition of disaccharides with the experimentally determined NAG2 structure at the acceptor site and biochemical data for mutated residues allowed us to propose that the interaction between Tyr145 and the +2 sugar moiety underpins acceptor sugar promiscuity. A glucose of β-1,6-glucan at the position +2 of the active site is predicted to have a steric impediment with Tyr145 explaining why this enzyme does not cross-link chitin to β-1,6-glucan.

The data presented here provide a platform for further studies towards the roles of the Crh enzymes, and their orthologues, in fungal biology. It may also be possible to exploit these enzymes as tools for generating longer homogenous and heterogeneous oligosaccharides with potential applications in biotechnology and glycochemistry.

## Methods

**Strains and growth conditions**. The *A. fumigatus* recipient strain *KU80ΔpyrG*[41] (a kind gift from Jean-Paul Latgé, Institut Pasteur, France), and *crh* mutants without *pyrG* were propagated at 37 °C on CM medium[42] supplemented with 5 mM uridine and 5 mM uracil. The *crh* mutants were maintained on CM. Conidia were prepared by propagating strains on solid medium for 48 h at 37 °C. Spores were harvested with 0.1% (v/v) Tween 20 in physiological saline, washed twice and re-suspended in sterile water. Conidial concentration was confirmed using a haemocytometer and viable cell counting. Mycelia sampled at specified times were harvested, washed with distilled water and frozen in liquid nitrogen. For DNA and RNA extraction the frozen mycelia were ground using a mortar and pestle. All plasmids were propagated in *E. coli* DH5α cells.

**Construction of Crh knockout mutants**. Due to limited selectable marker to choose and multiple *crh* genes in *A. fumigatus*, we designed a general knockout construct with a single but recyclable marker to generate multiple *crh* mutants. As the first step, P1 and P2 oligos were annealed and cloned into KpnI-SacI digested pBlueScriptII-SK to make construct I. This procedure replaced the KpnI and SacI sites with the MCS *HpaI-PacI-FseI-SmaI-NotI-AscI-HpaI*. The sites *PacI-FseI* are for upstream insertion; *NotI-AscI* for downstream insertion; *SmaI* between those pairs for cloning a reusable selectable marker and two *HpaI* sites beyond *PacI* and *AscI* for liberating a linear knockout fragment from the plasmid backbone. Then the marker neo-pyrG⁻neo was cut by *HpaI* from pCDA14 plasmid[43], gel extracted and cloned into the *SmaI* site and resulted in construct II containing the marker with pairs of unique sites for cloning the upstream and downstream regions from genomic DNA. This construct contains *pyrG* marker and two identical neo sequences for excision of the *pyrG* marker thus allowing multiple gene knockouts selection without marker switches. Finally, the 1–1.5 kb of upstream and downstream fragments of each *crh* gene were sequentially cloned by PCR from genomic DNA into *PacI-FseI* and *NotI-AscI* of construct II, respectively. Digesting the final construct with either *HpaI* or *PacI-AscI* released DNA for PEG-mediated protoplast transformation. Details of all primers are given in Supplemental Table 1. Apart from *PacI*, *FseI*, *AscI* and *NotI* were from NEB, other restriction enzymes and cloning reagents used in this paper were from Promega.

The recipient strain was *KU80ΔpyrG*⁻ and transformed colonies were screened on uridine/uracil autotrophy plates. PCR screening for all mutants was performed by three sets of primers to amplify *crh* genes, *pyrG* marker and neo to region after homologous downstream (neo-d-d). For mutants without *pyrG* marker upstream forward primer paired with downstream reverse primer were used to distinguish the mutants and the WT. Southern blot was conducted for all mutants after PCR diagnosis. Two probes were applied for each mutant: the 1.2 kb fragment of the marker and 1–1.5 kb of downstream or upstream. Genomic DNA of WT and mutants were digested by *XhoI*, *HindIII*, *PstI*, *BamHI-KpnI*, *BamHI* and *XhoI* for *pyrG*, *crh1* downstream, *crh2* downstream, *crh3* downstream, *crh4* upstream and *crh5* downstream probes, respectively. Visualisation was performed using the DIG DNA detection kit (Roche) according to the manufacturer's instruction.

**Incorporation of fluorescent oligosaccharides in *A. fumigatus***. Overnight culture of MM containing 3.75 μM SR, L5-SR, CH5-SR or NAG6-FITC (synthesised in lab) and 10⁵ conidia in a total volume of 400 μl were incubated in 24-well plate containing coverslips. After 16 h dark incubation at 37 °C cells were fixed in 3.7% paraformaldehyde with 0.1% Triton X-100 for 30 min. Following three PBS washes cells were stained by 10 μg ml⁻¹ DAPI for 20 min in the dark. After PBS wash 10 μg

ml[-1] CFW was used for staining the cell wall for 5 min. Finally, coverslips were put into fluorescent antifade mounting medium and sealed by nail polish before being subjected to fluorescence microscopy (Zeiss).

**Cloning, expression and purification of AfCrh5.** The bacterial expression vector pGEX6P1 (GE healthcare), which provides an N-terminal GST tag followed by a PreScission protease (PP) site, was modified by site-directed mutagenesis to include a six His-tag between the PP site and the *Bam*HI site. All *crh* genes were obtained from *A. fumigatus* ku80 cDNA by PCR using primers listed in Supplementary Table 1 and cloned into the modified expression vector. A truncated version of the *Af*Crh5 gene lacking the signal peptide and the GPI anchor (residues 22–275) was subcloned into the vector using *Bam*HI-*Not*I sites. The insert was confirmed by DNA sequencing. After transforming into *E. coli* BL21 (DE3) pLysS cells, GST-PP-His-fusion protein was expressed in autoinduction media at 18 °C for 40 h. Following GST beads purification, GST tag cleavage by PP overnight and clean-up through Superdex 75 column pure *Af*Crh5 was obtained at a yield of 3.4 mg L[−1]. The expression condition and purification steps for mutated proteins (E119Q, D121N, E123Q, W207A, R203A, Y145A, F137A, W221A, T218A, S109A and N135A) were the same as above.

**Enzymatic assays for AfCrh5.** For detecting chitinase or transglycosylase activity a FACE assay[44] was used. 2.5 mM of NAG2-NAG5 oligosaccharides (Megazyme) without or with laminarioligosaccharides G3–G5 (Megazyme) were used as substrates in 50 μl reaction containing 25 μg of *Af*Crh5 protein in McIlvaine buffer pH 4.9 for 16 h at 37 °C. Reaction was stopped by adding three volumes of ice-cold ethanol. After precipitation and evaporation of ethanol the reaction products were labelled with 750 nmol ANTS and used in a FACE gel.

The fluorescence assay to measure transglycosylase activity of *Af*Crh5 was carried out as below. The incubation mixture contained 0.1% CM–chitin, 18 μM SR–oligosaccharides, 1.5 μg of recombinant protein and 50 mM citrate buffer (pH 3.3, pH 3.7, pH 4.3, pH 4.9, pH 5.8, pH 6.3 or pH 7.4) in a total volume of 20 μl. Reactions were carried out at 37 °C for 0, 30, 60 and 90 min. The reactions were stopped with 20 μl 40% (v/v) formic acid. 5 μl aliquots from the stopped mixture were spotted in triplicates on to a filter paper (Whatman 3 mm) template corresponding to a standard 96-well microtitration plate. After drying, the paper was washed for 8–16 h with three changes of 66% (v/v) ethanol. The washing removed unreacted label, whereas CM–chitin and the high-$M_r$ products of its reaction with the SR-labelled acceptors remained attached to the paper. The paper was dried, placed into a 96-well microtitration plate and the fluorescence was measured in a FluoStar Omega Reader (BMH Labtech) equipped with a fluorescent detector and filters with excitation wavelength at 540 ± 10 nm and emission wavelength at 570 ± 10 nm.

**Crystallisation, data collection and structure determination.** Concentrated *Af*Crh5 at 10 mg ml[-1] was used for crystallisations screens using the sitting drop method. Each drop containing 0.2 μl of protein and 0.2 μl of reservoir solution (mother liquor) was set up by Mosquito (TTP Labtech). Needle-like clustered crystals appeared after 3 days from condition of 0.1 M sodium acetate trihydrate pH 4.6, 2.0 M ammonium sulphate. Single crystals were obtained after seeding in optimised condition of 0.1 M sodium acetate trihydrate pH 4.0, 1.6 M ammonium sulphate and 21 mM ZnCl₂. Apo crystals were cryoprotected with glycerol while complexed crystals were soaked with various sugars at room temperature for 1 h before cryoprotected in 2.5 M sodium malonate for testing and synchrotron trips. Data were collected at the European Synchrotron Radiation Facility (ESRF) (Grenoble, France) at beamlines ID29 and ID30A-1 and processed with Imosflm[45]. The apo structure was solved by molecular replacement using MOLREP[46] with the modelled *Sc*Crh1 structure[12] as the search model, followed by WarpNtrace model building[47]. The complex structure was obtained by molecular replacement using the apo structure. In both cases, REFMAC[48] was used for further refinement and iterated with model building using COOT[49]. Images were produced by PyMol[50]. The atomic coordinates and structure factors of *Af*Crh5 have been deposited in the Protein Data Bank with accession code 6IBU and 6IBW.

**Reporting summary.** Further information on experimental design is available in the Nature Research Reporting Summary linked to this article.

## Data availability

The atomic coordinates and structure factors of *Af*Crh5 are available from Protein Data Bank with accession codes 6IBU and 6IBW. A reporting summary for this article is available as a Supplementary Information file. The source data underlying Fig. 2 and Supplementary Figs. 2, 3 and 7 are provided as a Source Data file. Other data and reagents that support the findings of this study are available from the corresponding author upon reasonable request.

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

## Acknowledgements

We thank the European Synchrotron Radiation Facility beamlines ID29 and ID30A-1, Grenoble, France. This work was funded by MRC Programme Grant (M004139) to D.M.F.v.A., Research Start-up Funding of Guangxi Academy of Sciences (2017YJJ025) and Guangxi Natural Science Foundation (2018GXNSFAA138012) to W.F., ARAID to R.H-G., Ministerio de Economía (MINECO) to both R.H-G. and J.A. (CTQ2013-44367-C2-2-P and BFU2016-75633-P to R.H-G., and BIO2016-79289-P to J.A.), DGA to R.H-G. (E34_R17), and Comunidad de Madrid (FSE, FEDER) (S2017/BMD-3691 InGEMICS-CM) to J.A. A.B.S. was supported by contract from Comunidad de Madrid (FSE, FEDER).

## Author contributions

W.F. and D.M.F.v.A conceived the study; W.F. performed all genetics experiments, expressed and purified all the proteins, solved the structure and conducted FACE assay; A.B.S., V.F. and J.A. performed the fluorescent assays; B.W. performed cell biology; A.T.F. performed molecular biology; W.F., S.G.B., J.A., R.H.-G and D.M.F.v.A. interpreted the data and W.F., J.A., R.H.-G and D.M.F.v.A. wrote the manuscript with input from all authors.
