## [Peer Review File · Nature Communications]

Reviewers' comments:

Reviewer #1 (Remarks to the Author):

The study is focused on the analysis of the CRH family of *A.fumigatus*. The crystallography part of the ms looks fine and the binding of CRH5 to chitin is well detailed (but this reviewer is not a specialist of XRay structures) while the biochemical section of the ms is very poor. In contrast to yeast, the CRH proteins do not have any morphogenetic role in *A.fumigatus*. If the function of these proteins would be responsible for the covalent linkages between chitin and glucans, this reviewer would have expected a strong phenotype of the mutants taking into account the high concentration of the chitin in *Aspergillus* and the importance of chitin for the hyphal morphology. However, the quintuple mutant does not show any altered morphotype and even sensitivity to CR. Moreover, the binding of the SR oligos seem located in the septa. It is strange to see that the septa are not affected in the quintuple mutant.

The major flaw of the ms is the analysis of the biochemical activity of the enzymes of this family. The data are very poor and in the entire ms there is not any evidence of the existence or the demonstration of a covalent linkage between glucan and chitin. And this is true in vivo as well as in vitro. All the information given are based on indirect evidence showing the binding of a labeled oligosaccharide to a carboxymethylated chitin (found where in the fungal cell wall where only insoluble chitin exists?!). I am also puzzled by an enzyme accepting so many different donors. To confirm the binding of the labeled oligosaccharides to the cell wall, it is needed to use a native oligosaccharide or an oligosaccharide labeled with another fluorochrome to demonstrate that this putative covalent linkage of the CRH protein does exist. Are the SR oligos crossing the cell wall or does the binding of the SR oligo occur on the surface of the cell wall? There is no control with the sulforhodamine alone. Does this SR cross the cell wall?

Did the authors take into account the putative cross reactivity with beta 1,3 glucan modifying transglycosidases (many in this fungus). L5 SR may be an acceptor also for these enzymes. The cell wall should be positive with this L5SR. Did the authors check this point?

NAG5 is an acceptor. Do the authors see transfer products with NAG 5 or NAG6 with or without L5 which is also an acceptor ?

There is no doubt that this CRH5 is a binding protein but there is no proof in this ms that these CRH proteins have a true transglycosylase activity. May be the authors should focus exclusively their ms on the "lectin" aspect and send it to more specialized carbohydrate journals.

Minor comments

In Fig S4, the chitinase activity is very low. Is it NAG5 or NAG6 as stated in the Mat and Methods Fig. 2 What is FU2000 ? % of the initial substrate : 10% ou 0.01% ?

In Table 1 express Vmax en pmol/s/ μ g of protein. How was Km calculated; in other words what range of substrate concentrations did the authors use ? Km does not look in adequation with the poor enzymatic activity of this protein

Only one of the CRH has been analysed. What about the activity of the other ones?

Reviewer #2 (Remarks to the Author):

The authors present a comprehensive study on the activities of a number of Crh transglycosylases showing, among other findings, that none of the enzymes are required for cell wall stability. As part of this study, the authors report the crystal structure of AfCrh5 to 2.7 Angstrom resolution. The structure of the enzyme was determined at a pH where it has been shown to be active.

Significantly, the authors report a number of experiments that give the structure an appropriate biological relevance, as well as shed valuable light on this class of enzyme. The major findings of the study are the first-ever structure of an enzyme in this class, and the strong suggestion of a

preferred mechanism.

My concern is the apparent disconnect between the overarching motivation for the study implied in the Introduction as well as the Conclusion, which is given as the investigation of what may be therapeutic targets in the life cycle of an opportunistic fungus of clinical importance. However, the the ultimate finding given starting on line 352 reads:

“Unlike Crh mutants from *S. cerevisiae* and *C. albicans* in which single and multiple knockout mutants displayed hypersensitivity to Congo red 5, morphological aberrations at the mother-bud neck 14 and avirulence in mouse models of infection 15, all Crh mutants in *A. fumigatus* displayed no difference in terms of growth rate, germination, sporulation and sensitivity to cell wall stress reagents when compared to the parental strain.”

While the study is thorough and does add significantly to our knowledge of these enzymes, the lack of direct or even potential clinical impact does undercut somewhat the novelty of the study.

Specific comments:

The resolution of the structure should be stated in the abstract.

The crystallographic determination appears to have been nicely done for the most part, although I do have a few concerns, suggestions and questions.

First, I am somewhat uncomfortable with an R_{sym} as high as 13% (over 90% in the highest resolution shell). The average intensity in the highest resolution shell is just 70% over 1-sigma background, which leads me to suspect that the resolution might be somewhat overstated. I would be curious to see what these statistics looked like at 2.8 or 2.9 Å.

Second, the average temperature factor of the co-factor/ligand is almost double that for the protein and more than double that for the water molecules. This suggests a conservative approach in the assignment of water molecules (which is always best), but more significantly that the occupancy of the substrates is much less than 1.0. I would like to see at least one stereoview of the electron density for the substrates in the supplemental figures and some discussion of these apparent low occupancies. For one, is this high average temperature factor observed for both donor and acceptor? For another, given that the donor sugar residues in subsites -3 and -4 have no direct interaction with the enzyme (lines 230/231), how does the electron density (and temperature factors) of the substrate in -3 and -4 subsites compare with those of the substrate residues in -1 and -2. The authors should strongly consider adjusting the occupancy of the substrates(s) in their refinements so to bring the temperature factors into line with the protein atoms with which they interact.

As well, the authors mention that the substrates were soaked into unliganded crystals. Given the size of the substrates (one is a tetrasaccharide), this might explain the relatively low occupancy. Was any attempt made to co-crystallize? If unliganded crystal exists (or crystals with only one substrate) why is this not also reported?

Further, while the authors do mention on line 315/316 that AfCrh5 shows no activity toward the candidate acceptor NAG4, the authors should make clear in the discussion of the structure why a reaction between the substrates is not taking place right in the crystal.

I also have some specific questions about the refinement, and Table 2 in particular:

p13-14, Table 2

The resolution limits should include values in parenthesis corresponding to the bounds of the highest resolution shell.

What is the number of molecules in the asymmetric unit, Z?

Where is Rsym defined?

Rpim should be supplied with Rsym, and defined.

What fraction of data was sequestered in the calculation of Rfree?

The PDB accession code should be listed in the Table.

Finally, on the whole the manuscript was nicely written with few errors, I have some only minor typographical comments:

The spelling "sulfate" is IUPAC, but as this is a British journal, perhaps "sulphate" is more appropriate ...

Abstract:

The entirety of "in vitro" should be italicized.

p7, line 115

The sentence "Two cysteines pointed by light green forms a disulfide bridge." is awkward and has a tense mismatch. Might read better as "The two cysteines indicated by light green markers below the sequence form a disulfide bridge."

p7, line 118

"in vitro" was italicized earlier, and should be italicized here (like it is on line 118).

Reviewer #3 (Remarks to the Author):

The manuscript "Mechanisms of redundancy and specificity of the *Aspergillus fumigatus* Crh transglycosylases" describes a group of well-designed experiments looking at the Crh enzymes involved in cross-linking chitin to chitin and chitin to beta-1,3-glucan in *Aspergillus fumigatus*. The authors demonstrate that the *Aspergillus* enzymes are not required for viability and resistance to cell wall perturbation reagents, which is in contrast to the situation in yeasts. Using a genetic approach, the authors demonstrate the enzymes can cross link attach labeled chitin and labeled glucans into the cell wall. The authors expressed a recombinant enzyme and characterized the activity of the wild type enzymes and a number of mutant form enzymes in which key amino acids were replaced by alanine residues. They also characterized the crystal structure of the recombinant enzyme with substrate and acceptor oligosaccharides in place. They demonstrate that the enzyme structure would be able to accommodate for chitin and beta-1,3-glucan in the acceptor site.

The results presented by the authors are convincing and would be of interest to the wider research community. I would recommend that the article be accepted for publication. There are a number of suggestions I would make that I think will improve the manuscripts. These are:

1) Within the results and discussion section and the concluding remarks the authors discuss the importance of having the enzyme not function as a hydrolase by excluding water from the active site. The mention the facts that the cleavage product could occlude the site and that the acceptor site has some hydrophobic amino acids in it. The reaction mechanism requires that the cleaved chitin oligosaccharide in the acceptor site be replaced by the chitin or glucan acceptor. I would recommend that the authors discuss how this might be accomplished without introducing water into the acceptor site.

2) I find that the Materials and Methods section in the supplemental materials needs some

improvement. Specifically, the description of how the genes were cloned is hard to follow. Similarly, the description of how the recombinant gene was prepared is not described at all. Since others may want to repeat the analysis or use it as a model for a similar analysis, more complete descriptions of how the experiments were done is in order. Yet another problem I had with the Materials and Methods section was a lack of description of where the materials came from. For example, where did the authors get the various oligosaccharides used and where did the reagents used in the cloning and PCR reactions come from.

REPLY TO REFEREES' COMMENTS

All of the referees' original comments are shown in bold between quotation marks. Our responses are shown in plain text inserted into the full comments by the referees.

Referee #1 (Remarks to the Author):

“The study is focused on the analysis of the CRH family of *A.fumigatus*. The crystallography part of the ms looks fine and the binding of CRH5 to chitin is well detailed (but this reviewer is not a specialist of XRay structures) while the biochemical section of the ms is very poor.

In contrast to yeast, the CRH proteins do not have any morphogenetic role in *A.fumigatus*. If the function of these proteins would be responsible for the covalent linkages between chitin and glucans, this reviewer would have expected a strong phenotype of the mutants taking into account the high concentration of the chitin in *Aspergillus* and the importance of chitin for the hyphal morphology. However, the quintuple mutant does not show any altered morphotype and even sensitivity to CR. Moreover, the binding of the SR oligos seem located in the septa. It is strange to see that the septa are not affected in the quintuple mutant.”

Crh mutants in *S. cerevisiae* (Cabib *et al.*, 2007, *Mol Micro*) and *C. albicans* (Pardini *et al.*, 2006, *J. Biol. Chem*) displayed hyper-sensitivity at 50 µg/ml and 100 µg/ml CR respectively. In *A. fumigatus*, the Crh mutants (in particular the quintuple mutant) displayed slight sensitivity at 100 µg/ml and 200 µg/ml of CR. We have now included these data in Supplementary Figure 3, and updated the text accordingly. In our data we do not see effects on septa, in agreement with the non-lethal phenotype of the quintuple mutant.

“The major flaw of the ms is the analysis of the biochemical activity of the enzymes of this family. The data are very poor and in the entire ms there is not any evidence of the existence or the demonstration of a covalent linkage between glucan and chitin. And this is true in vivo as well as in vitro. All the information given are based on indirect evidence showing the binding of a labeled oligosaccharide to a carboxymethylated chitin (found where in the fungal cell wall where only insoluble chitin exists?!).”

In the publication describing the assay, the transglycosylation products generated by Crh1 with CM-chitin as donor and L5-SR as acceptor were characterised (by glycoside hydrolase sensitivity, TLC and mass spectrometry) as hybrid polymer molecules, composed of L5-SR and a portion of the donor polysaccharide attached to its non-reducing end (Mazan *et al.*, 2013, *Biochem J*, Fig.4, Table S1, Fig. S5 therein). We have generated the same products in our manuscript and therefore assume their identity is as per identified in this earlier publication. To further investigate this, we have conducted an additional experiment. Using FACE analysis with chitooligosaccharide and laminarioligosaccharide controls, we demonstrate that with either of these substrates alone, Crh5 is not able to produce longer, transglycosylated products, whereas it is when these substrates are used together. These data are included as Supplementary Figure 7a, 7b, 7c and 7d.

We were not able to formally confirm the structures of these longer products by MS, however, taken together these data support the proposed glucan:chitin transglycosylation activity of Crh5.

“I am also puzzled by an enzyme accepting so many different donors.”

Previous work (Mazan *et al.*, 2013, *Biochem J*) has shown that only the soluble chitin derivatives CM-chitin, glycol chitin and \geq NAG5 serve as the donors in the reaction. These are essentially the same structures – β (1,4)-linked *N*-acetylglucosamine polymers. However, these enzymes display considerable *acceptor* promiscuity, which we were puzzled by and drove us to investigate the molecular basis of this by solving the first crystal structure of the Crh family.

“To confirm the binding of the labeled oligosaccharides to the cell wall, it is needed to use a native oligosaccharide or an oligosaccharide labeled with another fluorochrome to demonstrate that this putative covalent linkage of the CRH protein does exist. Are the SR oligos crossing the cell wall or does the binding of the SR oligo occur on the surface of the cell wall? There is no control with the sulforhodamine alone. Does this SR cross the cell wall?”

We have previously shown that FITC labelled oligosaccharides such as FITC-CH4, FITC- β -1,3 and FITC- β -1,6 oligosaccharides could be incorporated into the cell wall in yeast (Fig. S2 in Mazan *et al.*, 2011, *Biochem.J.*). Specifically, free sulforhodamine did not react with Crh1 *in vitro* and did not label yeast cell walls *in vivo*. Furthermore, incorporation of labelled oligosaccharides is blocked in *crh1 Δ crh2 Δ* double mutant (Cabib *et al.*, 2008, *J.Biol.Chem.*; Blanco *et al.*, 2015, *FEBS J.*) and overexpression of wild type, but not inactive mutant of Crh1 or Crh2, in this strain recovered incorporation. Treatment of labelled cells with chitinase removed all the fluorescence (Cabib *et al.*, 2008, *J.Biol.Chem.*). We have now conducted additional experiments in *A. fumigatus* to explore this. In Supplementary Figure 4 and 5 we demonstrate that free sulforhodamine does not enter cells, and that NAG6 labeled with FITC instead of sulforhodamine is similarly incorporated.

“Did the authors take intoaccount the putative cross reactivity with beta 1,3 glucan modifying transglycosidases (many in this fungus). L5 SR may be an acceptor also for these enzymes. The cell wall should be positive with this L5SR. Did the authors check this point?”

The referee makes an interesting point. A possible experiment would be to use an *A. fumigatus* strain that is fully deficient of all Gel family genes. Unfortunately, as the referee points out, this is a large family (7 members) and a total knockout strain is not available. As an alternative, we point to an earlier publication exploring this in yeast (Cabib *et al.*, 2008, *J.Biol.Chem.*). The yeast *gas1 Δ* mutant showed enhanced fluorescence with SR-oligosaccharides, which may be accounted for by increased expression of Crh1 and the high chitin cell wall content in this mutant. Incorporation was blocked in a *gas1 Δ crh1 Δ crh2 Δ* triple mutant and could be restored by expression of CRH2.

“NAG5 is an acceptor. Do the authors see transfer products with NAG 5 or NAG6 with or without L5 which is also an acceptor ?”

To investigate the referee's question, we have conducted an additional experiment. Using FACE analysis with chitooligosaccharide and laminarioligosaccharide controls, we demonstrate that at low concentrations with either of these substrates alone, Crh5 is not able to produce longer, transglycosylated products, whereas it is when these substrates are used together. At high concentrations of products, a degree of transglycosylation with NAG5 as a substrate alone is apparent. These data are included as Supplementary Figure 7c.

“There is no doubt that this CRH5 is a binding protein but there is no proof in this ms that these CRH proteins have a true transglycosylase activity. May be the authors should focus exclusively their ms on the “lectin” aspect and send it to more specialized carbohydrate journals. “

The definition of an enzyme is that of a biological catalyst that enhances reaction rates that are otherwise low/absent in the absence of the catalyst. There is substantial evidence in the manuscript to suggest Crh5 is an enzyme. The active site as revealed by our structural analysis shows that of an active glycoside hydrolase/transglycosylase enzyme. We demonstrate hydrolysis/transglycosylation activity with two different assays – activity that is lost when we make structure-guided point mutants in the enzyme that target the catalytic machinery. We therefore do not agree that this is just a carbohydrate binding protein.

“Minor comments

In Fig S4, the chitinase activity is very low. Is it NAG5 or NAG6 as stated in the Mat and Methods”

The relatively low chitinase activity is in agreement with the transglycosylase function of these enzymes. Hydrolysis would essentially be an unproductive transglycosylation onto a water molecule – exactly the type of side reaction that the enzyme needs to avoid, and we have proposed a molecular mechanism underpinning this on the basis of our structural data. NAG6 was used to demonstrate the sugar standards (Supplementary Figure 7a) but not used in the subsequent activity assay.

“Fig. 2 What is FU2000 ? % of the initial substrate : 10% ou 0.01% ?”

F.U. represents arbitrary fluorescence units, this is now explained in the figure legend. F.U. values were obtained by measuring the fluorescence of the filter papers with a FLUOStar Omega Reader (BMH Labtech). Arbitrary fluorescence units at the time of adding the enzyme (time 0) were subtracted for each time point analysed in the reaction. As detailed in the Materials and Methods, to stop the reaction 20 µl of each reaction was with 20 µl of formic acid and then 5 µl was spotted onto the paper. So, F.U. values for each spot correspond to 12.5 % of the initial volume of the reaction.

“In Table 1 express Vmax en pmol/s/μg of protein. How was Km calculated; in other words what range of substrate concentrations did the authors use ? Km does not look in adequation with the poor enzymatic activity of this protein”

Kinetic parameters K_m , k_{cat} and k_{cat}/K_m were obtained using L5-SR and CH5-SR as acceptors. These values showed that CH5-SR gave higher binding to AfCrh5 than L5-SR, in agreement with the previously published data for yeast Crh proteins. The catalytic properties of the enzyme were determined using 1.5 μg of the enzyme with saturating concentrations of CM-chitin and different concentrations (2 μM, 4 μM, 8 μM, 16 μM, 32 μM) of acceptor at 37 °C in pH 4.9 buffer. Reactions time were such that steady state kinetics applied.

“Only one of the CRH has been analysed. What about the activity of the other ones?”

The referee is right to point out that we have only studied a single member of the Crh family. Prior to this work we have tried extensively to express/purify recombinant protein of other family members to a quantity/degree of purity suitable for enzymology and structural work, but were unsuccessful. However, the sequence conservation (Supplementary Figure 1) evaluated in the context of our structural data shows that the active sites of these enzymes are nearly identical, and we thus expect the principles we establish for the Crh5 enzyme to apply across the family.

Reviewer #2 (Remarks to the Author):

“The authors present a comprehensive study on the activities of a number of Crh transglycosylases showing, among other findings, that none of the enzymes are required for cell wall stability. As part of this study, the authors report the crystal structure of AfCrh5 to 2.7 Angstrom resolution. The structure of the enzyme was determined at a pH where it has been shown to be active.

Significantly, the authors report a number of experiments that give the structure an appropriate biological relevance, as well as shed valuable light on this class of enzyme. The major findings of the study are the first-ever structure of an enzyme in this class, and the strong suggestion of a preferred mechanism.

My concern is the apparent disconnect between the overarching motivation for the study implied in the Introduction as well as the Conclusion, which is given as the investigation of what may be therapeutic targets in the life cycle of an opportunistic fungus of clinical importance. However, the the ultimate finding given starting on line 352 reads:

“Unlike Crh mutants from *S. cerevisiae* and *C. albicans* in which single and multiple knockout mutants displayed hypersensitivity to Congo red 5, morphological aberrations at the mother-bud neck 14 and avirulence in mouse models of infection 15, all Crh mutants in *A. fumigatus* displayed no

difference in terms of growth rate, germination, sporulation and sensitivity to cell wall stress reagents when compared to the parental strain.”

While the study is thorough and does add significantly to our knowledge of these enzymes, the lack of direct or even potential clinical impact does undercut somewhat the novelty of the study.”

We appreciate that in the context of our original aims, our work somewhat “underdelivers”. Based on previous research in *S. cerevisiae* and *C. albicans* we hypothesized that genetic disruption of the Crh transglycosylases in *A. fumigatus* would cause significant cell wall integrity, morphogenesis and growth defects. Although we only saw those to a limited extent, it is quite possible that the Crh family may be important for pathogenesis during infection in a host. These enzymes are mostly located at the surface of the cell wall, the first point of contact with immune cells of the host - indeed they induce the production of specific antibodies (Arroyo *et al.*, 2007), trigger CD4⁺ T_H1 cells (Schutte *et al.*, 2009) and directly bind to cytokines such as IL-17A to facilitate survival (Zelante *et al.*, 2012). This is now more extensively discussed in the revised manuscript. However, this was not the only aim of our work – this is essentially an almost unexplored class of enzymes with limited knowledge of their function, structure and reaction mechanism and this is what we have studied in this manuscript.

“Specific comments:

The resolution of the structure should be stated in the abstract.”

We have now included this information in abstract.

“The crystallographic determination appears to have been nicely done for the most part, although I do have a few concerns, suggestions and questions.

First, I am somewhat uncomfortable with an R_{sym} as high as 13% (over 90% in the highest resolution shell). The average intensity in the highest resolution shell is just 70% over 1-sigma background, which leads me to suspect that the resolution might be somewhat overstated. I would be curious to see what these statistics looked like at 2.8 or 2.9 Å.”

In response to the referee we have scaled the data to 2.8 Å and have updated the reported statistics and structure refinement accordingly.

“Second, the average temperature factor of the co-factor/ligand is almost double that for the protein and more than double that for the water molecules. This suggests a conservative approach in the assignment of water molecules (which is always best), but more significantly that the occupancy of the substrates is much less than 1.0. I would like to see at least one stereoview of the electron density for the substrates in the supplemental figures and some discussion of these apparent low occupancies. For one, is this high average temperature factor observed for both donor and

acceptor? For another, given that the donor sugar residues in subsites -3 and -4 have no direct interaction with the enzyme (lines 230/231), how does the electron density (and temperature factors) of the substrate in -3 and -4 subsites compare with those of the substrate residues in -1 and -2. The authors should strongly consider adjusting the occupancy of the substrates(s) in their refinements so to bring the temperature factors into line with the protein atoms with which they interact.”

The referee points out a discrepancy between the temperature factors of the ligand and those of surrounding protein atoms and water molecules. Following the referee’s suggestion, we have included a stereo view of the omit map obtained after 20 cycles of rigid body refinement in the absence ligand atoms. This map (Supplementary Figure 6) reveals weaker electron density for the donor ligand in subsites -4 and -3. As the referee suggests and using the omit maps as a reference, we have divided the donor ligand into 3 independent ligands (positions -4, -3 and the disaccharide covering -2 to -1) for independent calculation of occupancies. All data and images are now included in Supplementary Figure 6 and main text has been modified accordingly.

“As well, the authors mention that the substrates were soaked into unliganded crystals. Given the size of the substrates (one is a tetrasaccharide), this might explain the relatively low occupancy. Was any attempt made to co-crystallize? If unliganded crystal exists (or crystals with only one substrate) why is this not also reported?”

As the referee correctly guessed, we have performed very extensive co-crystallisation studies to capture substrate/product complexes, but only soaks provided interpretable electron density maps. Following the referee’s suggestion, we have now grown further crystals of the protein alone, collected diffraction data set to 2.25 Å, refined this structure and have included this in the manuscript.

“Further, while the authors do mention on line 315/316 that AfCrh5 shows no activity toward the candidate acceptor NAG4, the authors should make clear in the discussion of the structure why a reaction between the substrates is not taking place right in the crystal.”

In the structure, we observed electron density for NAG4 at the donor binding site and NAG2 at the acceptor binding site, however this structure is the result of soaks with NAG4 only. Our enzyme assays have established that the minimum donor required for transglycosylation is NAG5 (Supplementary Figure 7b), thus reactions are unlikely to have taken place in the crystal. Instead, we think that this density represents two $\beta(1,4)$ -linked *N*-acetylglucosamine sugars, with the additional two occupying the +3 and +4 subsites being disordered. This has been discussed in the revised version.

“I also have some specific questions about the refinement, and Table 2 in particular:

p13-14, Table 2

The resolution limits should include values in parenthesis corresponding to the bounds of the highest resolution shell.”

This has been added to Table 2.

“What is the number of molecules in the asymmetric unit, Z?

Each asymmetric unit contains two crystallographically independent protein molecules with a root-mean-square deviation (RMSD) of 0.3 Å for 213 C α atoms. This has been included in the revised manuscript.

“Where is R_{sym} defined?

R_{pim} should be supplied with R_{sym}, and defined.

What fraction of data was sequestered in the calculation of R_{free}?”

The revised version of Table 2 now includes definition of R_{sym}, lists R_{pim} (and its definition) and lists the number of reflections used for calculation of R_{free}.

“The PDB accession code should be listed in the Table.”

This has been updated in Table 2. We have also deposited and included the apo structure.

“Finally, on the whole the manuscript was nicely written with few errors, I have some only minor typographical comments:

The spelling “sulfate” is IUPAC, but as this is a British journal, perhaps “sulphate” is more appropriate ...”

We have changed this throughout the manuscript.

“Abstract:

The entirety of “in vitro” should be italicized.”

We have changed this throughout the manuscript.

“p7, line 115

The sentence “Two cysteines pointed by light green forms a disulfide bridge.” is awkward and has a

tense mismatch. Might read better as “The two cysteines indicated by light green markers below the sequence form a disulfide bridge.” “

This has been corrected as suggested.

“p7, line 118

“*in vitro*” was italicized earlier, and should be italicized here (like it is on line 118).”

This has been corrected as suggested.

Reviewer #3 (Remarks to the Author):

“The manuscript “Mechanisms of redundancy and specificity of the *Aspergillus fumigatus* Crh transglycosylases” describes a group of well-designed experiments looking at the Crh enzymes involved in cross-linking chitin to chitin and chitin to beta-1,3-glucan in *Aspergillus fumigatus*. The authors demonstrate that the *Aspergillus* enzymes are not required for viability and resistance to cell wall perturbation reagents, which is in contrast to the situation in yeasts. Using a genetic approach, the authors demonstrate the enzymes can cross link attach labeled chitin and labeled glucans into the cell wall. The authors expressed a recombinant enzyme and characterized the activity of the wild type enzymes and a number of mutant form enzymes in which key amino acids were replaced by alanine residues. They also characterized the crystal structure of the recombinant enzyme with substrate and acceptor oligosaccharides in place. They demonstrate that the enzyme structure would be able to

accommodate for chitin and beta-1,3-glucan in the acceptor site.

The results presented by the authors are convincing and would be of interest to the wider research community. I would recommend that the article be accepted for publication. There are a number of suggestions I would make that I think will improve the manuscripts. These are:

1) Within the results and discussion section and the concluding remarks the authors discuss the importance of having the enzyme not function as a hydrolase by excluding water from the active site. The mention the facts that the cleavage product could occlude the site and that the acceptor site has some hydrophobic amino acids in it. The reaction mechanism requires that the cleaved chitin oligosaccharide in the acceptor site be replaced by the chitin or glucan acceptor. I would recommend that the authors discuss how this might be accomplished without introducing water into the acceptor site.”

Glycoside hydrolases often possess a degree of processivity involving a continued association of a long oligo/polysaccharide with the enzyme’s substrate binding side, re-“sliding” the longest product of an initial reaction over the active site to re-form productive binding of a substrate, followed by hydrolysis and so on.

While we do not have evidence of such a sliding mechanism here, it is possible that the critical solvent exposed tyrosine and tryptophan identified from our work form such a “slide”, allowing seamless deposition of a new acceptor, while shifting out the product of the first reaction step, without it fully dissociating from the active site first. Although we feel this is speculative, we have included a sentence in the discussion proposing this a possible mechanism.

“2) I find that the Materials and Methods section in the supplemental materials needs some improvement. Specifically, the description of how the genes were cloned is hard to follow. Similarly, the description of how the recombinant gene was prepared is not described at all. Since others may want to repeat the analysis or use it as a model for a similar analysis, more complete descriptions of how the experiments were done is in order. Yet another problem I had with the Materials and Methods section was a lack of description of where the materials came from. For example, where did the authors get the various oligosaccharides used and where did the reagents used in the cloning and PCR reactions come from.”

We totally agree with the referee that the Materials and Methods should allow others to precisely repeat our experiments (indeed that was the whole purpose of scientific publication when it originated centuries ago). We apologise for this omission. All the details of gene cloning, protein expression, purification and suppliers of materials have now been added in the revised manuscript.

REVIEWERS' COMMENTS:

Reviewer #1 (Remarks to the Author):

The authors have added numerous supplementary data and informations in the ms to better characterize the activity of the enzyme. They did not however answer the major comment that this referee pointed out previously and especially do not show any biochemical data demonstrating that the enzyme is the transglycosylase responsible for the establishment of the linkages between chitin and glucan in *A.fumigatus* cell wall. The CRH family is a very interesting and puzzling family of enzymes but the activity suggested has not been demonstrated yet.

In the new version of the ms the mutants display now an increased sensitivity to Congo Red although very mild. It is strange the authors have missed this effect since it was supposed to be the phenotype to investigate the most carefully in CRH mutants (which code for Congo Red Hypersensitivity). The sensitivity to Congo Red of complemented strains is not however mentioned in the revised manuscript and must be added (especially when the complementation of the quintuple mutant with CRH4 does not seem to change much the sensitivity to CR). It is not needed to complement all the mutants but 2 single mutants would be nice.

No explanation has been given for the lack of phenotype of the mutant at the septum level since the fluorescence study suggested that the activity of the CRH proteins and accordingly, the establishment of the chitin-glucan linkages occurs at the septum level. The authors said it is in agreement with the non-lethal phenotype. Indeed! but if these putative CRH transglycosidases display an essential role associated to the remodeling of the polysaccharides of the septa, a phenotype must be seen at the level of the septum and vegetative growth. There are many examples in yeasts and molds that when the septum is affected, dramatic phenotypes are seen.

There is still not a single chemical analysis of the products neither any demonstration that there is a reduction in the amount of chitin-glucan linkages in the quintuple mutant. A lot of the answers of the authors only mention data obtained in yeasts (where by the way no definite chemical analysis of the linkages was performed for any product produced by the enzymes). The authors cannot say they have generated the same product like in yeast proving that the activity must be the same because they did not characterize the product!!! And there are many examples of orthologous proteins which have different functions in yeasts and molds.

All data (and even more now in the revised version) are proving that the chitin-chitin linkage is favored over the chitin- glucan linkage. This is shown in various figures with oligosaccharides as well as with CM Chitin. The homotransglycosidase activity is even suggested by the authors and this would make more sense than the putative chitin- glucan transglycosylase. In fact this enzyme looks more like a chitinase which have some transglycosidase activity when the enzyme and the substrate do not encounter the right environmental conditions (which would explain the lack of specificity of the donor/acceptor substrates). The FACE patterns show that indeed the Crh5p has a chitinolytic activity. Moreover, if Crh5p was a true transglycosylase, the processing should not stop after the addition of one or two NAG. The CRH activity may be only responsible for the elongation of chitin in a way similar to the GEL/GAS family that cleave and elongate beta 1,3 glucans. This reviewer can accept also the possibility that the chemical environment of CRH5p used to study the enzymatic activity in vitro may not be the most appropriate to properly characterize the activity. If this is the case, the authors should look at the in vivo activity which means quantify the chitin-glucan linkages in the cell wall of the quintuple mutant and demonstrate that these linkages are absent. Unfortunately, such data have not been generated.

Reviewer #2 (Remarks to the Author):

The authors have responded to all of the comments that I made, including the comments on the significance of the determination. I have no further major comments, and only a few minor comments and suggestions. I would urge the authors to proof-read their manuscript closely for tense mismatches.

line 15: I'm not sure about using the first person in an abstract, but it should be "we solved".

line 27: Change "between" to "among".

line 31: Change "between" to "among".

line 107: Delete "only".

line 126: "No growth rate"? If this means "No growth rate defects", maybe move "defects" to the start of the sentence: "No defects in growth rate, germination or sporulation were detected ..."

line 233: Change "belonged to" to "belong to".

line 234: "No conformational were observed between ..." There's a word missing here.

line 278: Change "were chitin" to "was chitin".

line 341: Change "mutation of it" to "its mutation".

line 355: Change "density" to "electron density".

line 403: Change "to comprehensively understand" to "to give a comprehensive understanding of"

Responses to the reviewers:

Reviewer #1 (Remarks to the Author):

The authors have added numerous supplementary data and informations in the ms to better characterize the activity of the enzyme. They did not however answer the major comment that this referee pointed out previously and especially do not show any biochemical data demonstrating that the enzyme is the transglycosylase responsible for the establishment of the linkages between chitin and glucan in *A.fumigatus* cell wall. The CRH family is a very interesting and puzzling family of enzymes but the activity suggested has not been demonstrated yet.

There is still not a single chemical analysis of the products neither any demonstration that there is a reduction in the amount of chitin-glucan linkages in the quintuple mutant. A lot of the answers of the authors only mention data obtained in yeasts (where by the way no definite chemical analysis of the linkages was performed for any product produced by the enzymes). The authors cannot say they have generated the same product like in yeast proving that the activity must be the same because they did not characterize the product!!! And there are many examples of orthologous proteins which have different functions in yeasts and molds. All data (and even more now in the revised version) are proving that the chitin-chitin linkage is favored over the chitin- glucan linkage. This is shown in various figures with oligosaccharides as well as with CM Chitin. The homotransglycosidase activity is even suggested by the authors and this would make more sense than the putative chitin-glucan transglycosylase. In fact this enzyme looks more like a chitinase which have some transglycosidase activity when the enzyme and the substrate do not encounter the right environmental conditions (which would explain the lack of specificity of the donor/acceptor substrates). The FACE patterns show that indeed the Crh5p has a chitinolytic activity. Moreover, if Crh5p was a true transglycosylase, the processing should not stop after the addition of one or two NAG. The CRH activity may be only responsible for the elongation of chitin in a way similar to the GEL/GAS family that cleave and elongate beta 1,3 glucans. This reviewer can accept also the possibility that the chemical environment of CRH5p used to study the enzymatic activity *in vitro* may not be the most appropriate to properly characterize the activity. If this is the case, the authors should look at the *in vivo* activity which means quantify the chitin-glucan linkages in the cell wall of the quintuple mutant and demonstrate that these linkages are absent. Unfortunately, such data have not been generated.

Author response:

While we have not quantified the different fractions of chitin in isolated cell walls of the quintuple mutant as definitive proof for the biological function of these proteins in chitin-glucan and/or chitin-chitin linkages, there are several results in the manuscript showing that both *in vitro* and *in vivo* AfCrh5 enzyme is a transglycosylase. *In vivo* we demonstrated that Crh proteins are responsible for the incorporation of laminarioligosaccharides and chito-oligosaccharides, the latter ones more efficiently, suggesting their involvement in chitin transglycosylation. *In vitro*, AfCrh5 behaves like a chitin transglycosylase using soluble carboxymethyl chitin or chito-oligosaccharides (\geq NAG5) as glycosyl donors and preferentially oligosaccharides derived from chitin but

also those derived from β -1,3 glucan as acceptors, both in the fluorescence assay and in the FACE assay where very faint bands of higher oligomers formed by incubation of AfCrh5 with NAG5 alone or NAG5 with G4 or G5. All transglycosylases have intrinsic chitinase activity as this is the first step of the reaction mechanism. Crh5 it is primarily a transglycosylase because it is able to transglycosylate at very low concentrations of substrates (micromolar) whereas glycosidases typically catalyse transglycosylation as a reverse hydrolysis, at millimolar concentrations of substrates. We have now discussed this in the revised manuscript.

We agree with the reviewer that most of the data would be in agreement with the idea that the chitin-chitin linkage is favored over the chitin-glucan linkage. Thus, we have included more comments throughout the revised manuscript and in particular in the discussion about the preference of the AfCrh5 enzyme for chitooligosaccharides as glycosyl acceptors and therefore its possible involvement on chitin-chitin linkages.

In the new version of the ms the mutants display now an increased sensitivity to Congo Red although very mild. It is strange the authors have missed this effect since it was supposed to be the phenotype to investigate the most carefully in CRH mutants (which code for Congo Red Hypersensitivity). The sensitivity to Congo Red of complemented strains is not however mentioned in the revised manuscript and must be added (especially when the complementation of the quintuple mutant with CRH4 does not seem to change much the sensitivity to CR). It is not needed to complement all the mutants but 2 single mutants would be nice.

Author response:

When we analysed *crh* mutants, sensitivity to 200 μ g/ml of CR we realized that all mutants showed some sensitivity to high concentrations of this compound. Moreover this sensitivity was slightly increased in the quintuple mutant and restored by complementation with *crh4* (Supplementary Fig. 3), since the quintuple mutant transformed with *crh4* gene show the same sensitivity as the quadruple *crh1 crh2 crh3 crh5* mutant. As suggested by the reviewer we have described these results in the new version and discuss in more detail in the context of a redundant function for Crh proteins in cell wall assembly.

No explanation has been given for the lack of phenotype of the mutant at the septum level since the fluorescence study suggested that the activity of the CRH proteins and accordingly, the establishment of the chitin-glucan linkages occurs at the septum level. The authors said it is in agreement with the non-lethal phenotype. Indeed! but if these putative CRH transglycosidases display an essential role associated to the remodeling of the polysaccharides of the septa, a phenotype must be seen at the level of the septum and vegetative growth. There are many examples in yeasts and molds that when the septum is affected, dramatic phenotypes are seen.

Author response:

Our results show that *in vivo* SR-laminarioligosaccharides and SR-chitooligosaccharides (the latter more efficiently) are incorporated into the lateral cell wall and particularly

at the septa, thus functioning as artificial acceptors. This incorporation is blocked when the five *crh* *A. fumigatus* genes are deleted simultaneously. Moreover, expression of either *crh3* or *crh4* genes restores the formation of the crosslinks in the quintuple mutant, suggesting that the five *crh* genes provide the redundancy of chitin cross-linking at his localization. We have no explanation for the absence of phenotype at the septum but in yeast, where Crh proteins are clearly involved in the control bud-neck growth through the formation of covalent crosslinks between chitin and glucan, the *crh1 crh2* double mutant also does not present a morphogenic phenotype. In that case, since two structures, the chitin ring and the septin ring control yeast morphogenesis at this region, the morphogenetic phenotype is only observed in a *cla4 crh1 crh2* strain. Thus, clearly future work will be necessary to establish the functional role for these proteins at the septum in *A. fumigatus*. We have discussed this in the revised version.

Reviewer #2 (Remarks to the Author):

The authors have responded to all of the comments that I made, including the comments on the significance of the determination. I have no further major comments, and only a few minor comments and suggestions. I would urge the authors to proof-read their manuscript closely for tense mismatches.

line 15: I'm not sure about using the first person in an abstract, but it should be "we solved".

line 27: Change "between" to "among".

line 31: Change "between" to "among".

line 107: Delete "only".

line 126: "No growth rate"? If this means "No growth rate defects", maybe move "defects" to the start of the sentence: "No defects in growth rate, germination or sporulation were detected ..."

line 233: Change "belonged to" to "belong to".

line 234: "No conformational were observed between ..." There's a word missing here.

line 278: Change "were chitin" to "was chitin".

line 341: Change "mutation of it" to "its mutation".

line 355: Change "density" to "electron density".

line 403: Change "to comprehensively understand" to "to give a comprehensive understanding of"

Author response:

All the minor comments from the reviewer have been corrected in text. Many thanks for pointing these out.